# Identifying Dopamine D3 Receptor Ligands through Virtual Screening and Exploring the Binding Modes of Hit Compounds

**DOI:** 10.3390/molecules28020527

**Published:** 2023-01-05

**Authors:** Hongshan Jin, Chengjun Wu, Rui Su, Tiemin Sun, Xingzhou Li, Chun Guo

**Affiliations:** 1Key Laboratory of Structure-Based Drug Design and Discovery Ministry of Education, Department of Pharmaceutical Engineering, Shenyang Pharmaceutical University, Shenyang 110016, China; 2Beijing Institute of Pharmacology and Toxicology, Beijing 100850, China

**Keywords:** dopamine D3 receptor, virtual screening, induced-fit docking, binding pose metadynamics simulation, molecular dynamics

## Abstract

The dopamine D3 receptor (D3R) is an important central nervous system target for treating various neurological diseases. D3R antagonists modulate the improvement of psychostimulant addiction and relapse, while D3R agonists can enhance the response to dopaminergic stimulation and have potential applications in treating Parkinson’s disease, which highlights the importance of identifying novel D3R ligands. Therefore, we performed auto dock Vina-based virtual screening and D3R-binding-affinity assays to identify human D3R ligands with diverse structures. All molecules in the *ChemDiv* library (>1,500,000) were narrowed down to a final set of 37 molecules for the binding assays. Twenty-seven compounds exhibited over 50% inhibition of D3R at a concentration of 10 μM, and 23 compounds exhibited over 70% D3R inhibition at a concentration of 10 μM. Thirteen compounds exhibited over 80% inhibition of D3R at a concentration of 10 μM and the IC_50_ values were measured. The IC_50_ values of the five compounds with the highest D3R-inhibition rates ranged from 0.97 μM to 1.49 μM. These hit compounds exhibited good structural diversity, which prompted us to investigate their D3R-binding modes. After trial and error, we combined unbiased molecular dynamics simulation (MD) and molecular mechanics generalized Born surface area (MM/GBSA) binding free-energy calculations with the reported protein–ligand-binding pose prediction method using induced-fit docking (IFD) and binding pose metadynamics (BPMD) simulations into a self-consistent and computationally efficient method for predicting and verifying the binding poses of the hit ligands to D3R. Using this IFD-BPMD-MD-MM/GBSA method, we obtained more accurate and reliable D3R–ligand-binding poses than were obtained using the reported IFD-BPMD method. This IFD-BPMD-MD-MM/GBSA method provides a novel paradigm and reference for predicting and validating other protein–ligand binding poses.

## 1. Introduction

Dopamine receptors comprise a class of G-protein coupled receptors (GPCRs) that are important for the central nervous system (CNS). At least five dopamine receptor subtypes are produced, namely D1R, D2R, D3R, D4R, and D5R. D1R and D5R are members of D1-like dopamine receptor family, whereas D2R, D3R, and D4R are members of the D2-like family [1]. An imbalance between dopaminergic neurotransmission and dopamine receptors is the basis of many neurological and mental diseases, such as Parkinson’s disease, Huntington’s chorea, schizophrenia, and drug abuse [2].

The dopamine D3 receptor (D3R) is found within a key neuronal network involved in motivation and cognition and does not appear to regulate movements. Compared with other dopamine receptor subtypes, the D3R exhibits restricted distribution in the mesolimbic system and has the highest affinity for endogenous dopamine. Therefore, the D3R subtype is considered an important target for treating various nervous system diseases such as schizophrenia, Parkinson’s disease, and abuse. D3R antagonists can help improve psychostimulant addiction and relapse, whereas D3R agonists can enhance the response to dopaminergic stimulation and have potential applications for treating Parkinson’s disease [3].

Despite a plethora of evidence supporting an important role for D3R in neurological diseases, the results of drug-development efforts to target D3R have not been ideal. To our knowledge, only three D3R-preferred D3R/D2R agonists developed before 2005 (pramipexole, ropinirole, and rotigotine transdermal patches) have been marketed for treating Parkinson’s disease, restless legs syndrome (RLS), or other disorders [4,5,6]. However, these drugs were not originally developed for D3R. The development of D3R-selective antagonists (including GSK 598809 and ABT-925) as well as the D3R-preferring D3R/D2R antagonist S33138 (intended for treating schizophrenia and drug abuse) were all discontinued due to insufficient activities(Figure 1) [7]. A successful example of a D3R-targeted drug is cariprazine, a D3R-preferred D3R/D2R partial agonist launched in 2015 for the treating schizophrenia and bipolar disorder [8]. In addition, the D3R-preferring D3R/D2R antagonist F17464 demonstrated therapeutic efficacy in improving the symptoms of acute exacerbation of schizophrenia with a favorable safety profile in a Phase II clinical trial [9].

In this article, we searched for new D3R ligands by performing virtual screening, followed by experimental testing with the most promising compounds. We also investigated the binding modes of the obtained D3R ligands using a series of computational biology methods in order to provide more structure–activity-relationship information for designing novel D3R ligands.

## 2. Results and Discussion

### 2.1. Virtual Screening

#### 2.1.1. Receptor Selection and Preparation

PDB 3PBL is the only crystal structure reported for D3R to date [10]. In that structure, the D3R forms a complex with the D2R/D3R-specific antagonist eticlopride, revealing important characteristics of the ligand-binding pocket. Previous MD simulations showed that the protein had right geometry when complexed with the ligand, eticlopride [11]. Therefore, this crystal structure was suitable for virtual-screening research. Chain A of PDB 3PBL has a more complete structure than chain B; thus, chain A was selected as the receptor for virtual screening. Ligands, ions, and water were removed from chain A; Gasteiger charges were generated; and energy minimization was preformed using CHIMERA 1.5.3 software [12]. An interaction grid for AutoDock Vina was generated using the center of the original ligand (eticlopride) as the center of the grid. The grid spacing was set to 1 Å, and the grid dimensions were selected so as to include all atoms of the original ligand (eticlopride), which were then augmented by ±10 Å in x, y, and z directions.

#### 2.1.2. Validation of the Docking Method

To investigate the applicability and reliability of the virtual-screening method based on AutoDock Vina docking, we first attempted to re-dock the original ligand, eticlopride, to its binding site using AutoDock Vina. Molecular docking was carried out with global searching-exhaustiveness settings of 8, 56, and 400, which corresponded to short, medium, and long options, respectively. The re-docking results for eticlopride were consistent across all three search modes. The RMSD values of the poses generated with all three search modes (relative to that of the original pose) were 1.10 Å, 1.15 Å, and 2.11 Å respectively (Figure 2). Previous computational data suggested that a pose with an RMSD value of <2 Å is a good pose [13]. The results imply that, regardless of the docking mode (short, medium, or long), AutoDock Vina docking was capable of relatively accurate predictions of the binding poses of D3R ligands. Although increasing the searching exhaustiveness from short to long leads to a significant increase in the computational cost, it did not provide improved accuracy or precision. Thus, the short option with searching exhaustiveness of 8 was selected for subsequent virtual screening based on AutoDock Vina docking in this study.

#### 2.1.3. Virtual Screening via AutoDock Vina Docking and Compounds Selection

The molecules in the ChemDiv compound database (consisting of 1,535,478 compounds at the time this study was initiated) were analyzed by molecular docking using the AutoDock Vina package in short-searching mode. The top-ranking 300 compounds with the lowest AutoDock Vina score (docking energies) were retained for analysis. They were divided into 40 clusters, and the compound with the lowest AutoDock Vina score in each cluster was chosen for further analysis. Finally, 37 compounds commercially available were purchased for biological evaluations. The structures and AutoDock Vina scores of the selected compounds are shown in Figure 2. The docking scores of the active compounds are also listed in Table 1.

### 2.2. D3R Binding-Affinity Assays

All 37 compounds were tested for their D3R-binding affinities at 10 µM in competitive-binding assay using [^3^H]-spiperone [11], with the D3R-preferred D3R/D2R agonist BP897 as positive control. The assay results showed that our virtual screening had a high hit rate. Among these 37 compounds tested, 27 compounds exhibited over 50% inhibition of D3R at a concentration of 10 μM, and 23 compounds exhibited over 70% D3R inhibition at a concentration of 10 μM. The inhibition data and AutoDock Vina scores of hit compounds are shown in Table 1. Thirteen compounds exhibited over 80% inhibition of D3R at a concentration of 10 μM. The D3R IC_50_ values of the compounds with the 80% inhibition rates were determined by generating concentration-displacement curves. Their IC50 values for inhibiting D3R binding ranged from 0.97 μM to 5.98 μM (Table 1). The best five active compound structures are shown in Figure 3. In each assay, the positive control, BP897, showed ≥98% inhibition for D3R at 10 µM, which confirmed the reliability of the D3R binding-affinity assays.

### 2.3. Exploring the D3R-Binding Modes of the Hit Compounds 

#### 2.3.1. Analyzing the AutoDock Vina-Based Docking Poses of the Hit Compounds

The five compounds were selected as hit compounds with the IC_50_ between 0.97 μM to 1.49 μM(Figure 3). Our biochemical analysis confirmed that the hit compounds indeed tightly bound to D3R. Moreover, the hit compounds exhibited good structural diversity, which made us very interested in their D3R-binding modes. Thus, we explored the binding modes of the AutoDock Vina docking poses of the hit compounds to D3R and generated corresponding protein–ligand-interaction diagrams (Figure 4). The ΔG binding values of D638-0102, D280-0447, E776-0059, F072-0905, and L227-1012 to D3R (calculated using the MM/GBSA method) were −46.74, −44.82, −70.76, −47.53, and −72.56 kcal/mol, respectively.

In view of the obvious similarity between the docking posture of AutoDock Vina and the binding mode of the hit compounds, we are not sure to what extent the docking posture of AutoDock Vina reflects the actual ligand posture when it binds to D3R. In addition, the absolute value of the ΔG binding value of AutoDock Vina docking posture is not high. This is due to the fact that the docking of AutoDock Vina does not take into account the flexibility of the receptor [10], which indicates that the docking posture of AutoDock Vina may be inconsistent with the actual ligand posture. On the side, like other docking scoring functions, AutoDock Vina scoring cannot completely and accurately distinguish the correct posture from the assumed candidate posture generated by the docking package [14].

#### 2.3.2. Methods for Inferring the Binding Modes of Protein–Ligand Complexes Using Computational Methods

It is a challenging task to determine the exact binding position of ligand and protein by calculation method. To predict the correct binding pose of a ligand with a protein, some docking methods have been developed that consider the flexibility of both the ligand and receptor, such as the IFD [15], GOLD [16], and flexible docking [17] methods. Among these flexible docking methods, IFD approach (developed by Schrödinger, Inc. New York City, NY, USA) is designed for simultaneous conformational sampling of both the receptor and the ligand [18]. The IFD protocol generates a constrained minimization of the receptor, followed by initial glide docking of the ligands using a softened potential. A select set of the docked poses are passed on to prime for a refinement step. After a prime side-chain prediction and minimization, the best receptor structures for each ligand are passed back to glide for re-docking of the ligand, and then, the binding energies for each output pose are estimated as the IFD score [19]. IFD usually identifies a structure with good accuracy within the top 5–10 results. Although the IFD method has been used with some success, its lack of reliability is its obvious disadvantage. Specifically, robustly ascertaining the correct structure from the many possibilities generated using IFD can be difficult [20]. 

BPMD analysis is an automated, enhanced-sampling, metadynamics-based protocol included in the Schrödinger software package. This protocol can be used to reliably discriminate between the correct ligand binding pose and plausible alternatives generated from docking studies [20]. Three BPMD scores (PoseScore, PersScore, and CompScore) were used to assess the ligand-binding stabilities. The PoseScore is indicative of the average RMSD from the starting pose. The steepest increase of this value indicates conditions where ligand binding to the protein is unstable. A PoseScore of <2 Å is considered stable for a complex ligand. The PersScore is a measure of the hydrogen bond (HB) persistence during a metadynamics simulation, with values ranging between 0 and 1, where higher values correspond to more stable complexes. The CompScore is a composite score obtained by linearly combining the PoseScore and PersScore, where lower values correspond to more stable complexes.

#### 2.3.3. Validating the AutoDock Vina Docking Poses

Initially, the AutoDock Vina docking poses of the hit compounds were analyzed by performing BPMD simulations to evaluate their reliabilities. As a control, and the binding pose of eticlopride in the protein crystal structure (PDB 3PBL) was also validated by performing BPMD simulations. 

The average RMSD values of the heavy atoms of ligands over 10 trials were plotted versus the simulation time (Figure 5). The maximum RMSD value of each curve was the PoseScore value of the corresponding pose. The PoseScore, PersScore, and CompScore values for each ligand are also presented in Figure 3. The PoseScores of the AutoDock Vina docking poses of our hit compounds were all greater than 2 Å. The PersScores showed that the HB-retention ratios during the simulations were also very low: only the HB-retention ratio of the pose for E776-0059 was relatively high (23.2%), whereas the HB-retention ratios of the other ligand poses were 0 or close to 0. Consequently, the CompScores of these poses were all above 1.5. These results suggest that the AutoDock Vina docking poses of the hit compounds were not stable at the active site and were unlikely the actual ligand poses. In contrast, the PoseScore, PersScore, and CompScore for the eticlopride/DD3R complex (PDB 3PBL) were 1.211 Å, 0.764, and −0.609, respectively, suggesting that the position and HB network of eticlopride were well-preserved during the BPMD simulations and that the eticlopride/DD3R complex was very stable.

#### 2.3.4. Exploring Potential Binding Poses Using a Combination of IFD and BPMD Simulations 

Based on the previous literature [14,15,20], we used the combination of IFD and BPMD simulation to explore the potential binding poses of the hit compounds when bound to D3R (Figure 6). Using this method, we also studied the potential binding posture of eticloprost with D3R and compared the assumed correct posture with the correct posture of eticloprost in crystal structure (PDB 3PBL) to test the reliability of this method.

Using the LigProp scheme in Schrödinger software package, the promoters and isomers of eticlopride and the hit compounds at pH 7 ± 2 were generated for the first time. Next, eticlopride and the hit compounds were flexibly docked to D3R using IFD scheme to consider the potential conformational changes of the residues, with the aim of finding potential stable postures for these compounds. A set of poses was generated for each promoter and isomer of eticlopride and the hit compounds. Then, the IFD output gestures are clustered by using the structural interaction fingerprint (SIFt) contact similarity score to eliminate redundant gestures [21]. In each cluster, only the posture with the highest IFD score was retained. The numbers of poses and clusters generated for each compound by SIFt contract similarity score are shown in Table 2. According to Schrödinger Desmond [22], the first three non-redundant gestures ranked by IFD scores are selected for further BPMD simulation. These non-redundant poses were re-ranked by PoseScore, PersScore, and Compscore (Table 3). Because the Compscore affects posture maintenance and hemoglobin maintenance, we prefer to use Compscore as the main score for evaluating the stability of combined posture in this study. The posture with the lowest Compscore of each D3R ligand (the posture with the highest BPMD score) is determined to be the most likely binding posture of the ligand. Comparing the top BPMD-scored pose of eticlopride, obtained through this procedure, with the original binding poses for eticlopride in the D3R/eticlopride crystal complex (PDB 3PBL), we found that these two poses coincided well, with an RMSD of only 0.74 Å (Figure 7). This finding showed that the method of exploring the potential binding poses via a combination of IFD and BPM was relatively reliable. In addition, the PoseScores, PersScores, and Compscores of the posture with the highest BPM score of the hit compounds are obviously better than their corresponding docking postures of AutoDock Vina (Table 3 and Figure 3). These results showed that the attitude that hits the compound with the highest BPM score is more stable than their corresponding automatic docking posture of Vina.

#### 2.3.5. MD Analysis of BPMD-Output Poses

Complementing IFD with unbiased MD simulations is a straightforward solution in principle but not in practice due to the severe time-scale limitations of MD [23]. The solution for improving the simulation-time efficiency is to use enhanced sampling techniques. Metadynamics is a powerful technique for enhancing the sampling during MD simulations and reconstructing the free-energy surface as a function of using few selected degrees of freedom, often referred to as collective variables (CVs) [24].

Although the top BPMD-scored poses are more stable than the AutoDock Vina docking poses, these findings do not enough to prove that the top BPMD-scored poses are the actual binding poses. To further verify the top BPMD-scored poses obtained by IFD and BPMD simulations, we performed unbiased MD simulations for the ligand–D3R complexes containing these top BPMD-scored poses using Desmond. MD simulations were run for 100 ns using default simulation parameters, collecting structural data every 0.1 ns for a total of 1000 frames per complex. The RMSD values of the D3R protein Cα atoms and Lig Fit Prot of the hit ligands found during the MD simulations are presented in Appendix A. The simulation results showed that the RMSD values of protein Cα atoms and the Lig Fit Prot value reached equilibrium soon after the simulation was started and remained stable until the end of the simulation, and there was no case in which the Lig Fit Prot value was significantly greater than the corresponding protein RMSD value. These data confirmed that the D3R protein was stable during the simulations and that the ligands (the hit compounds) in the complexes did not diffuse out of their original binding sites; that is, the entire protein–ligand complexes were stable.

To estimate the most populated representative conformation in each MD simulation, trajectory-clustering analyses were conducted using the “RMSD-Based Clustering of Frames from Desmond Clustering Trajectory” in Maestro. The conformation with the most neighbors in the trajectory clusters was selected as the representative conformation for each protein–ligand complex. Next, the positions, ligand–protein interactions, and ΔGbind values of each ligand in these representative conformations were compared with that of the corresponding initial conformations of the MD simulations, that is, the ligand–D3R conformation containing the top BPMD-scored pose.

The representative conformation obtained through MD simulations (hereinafter referred to as the representative conformation) of each D3R–ligand complex was aligned with its corresponding ligand–D3R complex conformation containing the top BPMD-scored pose (hereinafter referred to as the top BPMD-scored conformation) using the Align Binding Sites task tool in the Schrödinger suite (Figure 5). The comparative results showed that the positions of ligands D638-0102, D280-0447, L227-1012, and E776-0059 in their representative conformations deviated slightly from the positions of their top BPMD-scored pose with RMSD values of 1.50 Å, 0.78 Å, 1.60 Å, and 2.18 Å, respectively. The position of E776-0059 in the representative conformation deviated considerably from the position of its top BPMD-scored pose, and the RMSD value reached 3.33 Å (Figure 8). These results showed that the positions of most ligands were well-preserved before and after unbiased MD simulations of their top BPMD-scored conformations although the position of E776-0059 clearly shifted.

The ligand–protein interactions for each ligand in their representative conformations and top BPMD-scored conformation were also carefully compared. The 2D schematic diagrams of the ligand–protein interactions of each ligand in different complex conformations were drawn using the Ligand Interaction Diagram Panel of Schrödinger software (Figure 9). These key amino acid residues predicted to form strong ligand–protein interactions (including HBs, π−π stackings, and ionic bridges) are listed in Table 2. Although many amino acid residues were involved in the ligand-binding site of D3R, only a few residues played a critical role in maintaining ligand–protein binding. These key amino acid residues of D3R included Asp110, Ser182, Ile183, Phe345, Hie349, Hie349, and occasionally Phe346 and Asn352. Among these key amino acid residues, Asp110 was the most critical. It could form HBs (and sometimes ion bridges) with all five ligands studied regardless of whether the ligand was in its top BPMD-scored pose or in its representative conformation. The roles of Hie349 and Phe345 were also important. In the top BPMD-scored conformations of E776-0059 and F072-0950, Hie349 formed π−π stackings with both E776-0059 and F072-0950, whereas in the representative conformations, Hie349 formed one or two π−π stackings with each of four different ligands (D638-0102, D280-0447, E776-0059, and F072-0950). In addition, Hie349 could also form an HB with F072-0950 as the acceptor. In the top BPMD-scored conformations, Phe345 formed π−π stackings with D638-0102, E776-0059, and F072-0950, whereas in the representative conformations, Phe345 formed one or two π−π stackings with each of three different ligands (L227-1012, E776-0059, and F072-0950). Tyr365 was also found to be critical for ligand binding. In the top-BPMD scored conformations, Tyr365 formed π−π stacking and HBs with D638-0102 and D280-0447, respectively, whereas in the representative conformations, Tyr365 showed π−π stacking with D638-0102, π−π stacking and an HB with L227-1012, and an HB with both E776-0059 and F072-0950. Residues Ser182 and Ile 183 also played important roles in maintaining ligand and protein functions. Both Ser182 and Ile 183 could form HBs with ligands. In the top BPMD-scored conformation, Ser182 acted as both a proton acceptor and donor to form two HBs with D638-0102 and as a proton donor to form HBs with L227-1012, E776-0059, and F072-0950. However, in the representative conformation, Ser182 could only form one HB with D638-0102. These results suggest that Ser182 preferentially forms HBs with ligands in their top BPMD-scored pose. In contrast, Ile183 did not show this preference for ligand poses, and it formed HBs with E776-0059 and F072-0950 in both the top BPMD-scored conformations and representative conformations. The other two amino acid residues, Phe346 and Asn352, occasionally showed π−π stacking or HBs with ligands.

Analyzing these ligand–protein interactions when the ligands were in different conformations (Table 4) revealed that after unbiased MD simulations, A greater number of strong interactions (such as HBs, ionic bridges, and π−π stacking) tend to form between the ligands and key amino acid residues of D3R. In the top BPMD-scored conformations, the ligands D638-0102, D280-0447, L227-1012, E776-0059, and F072-0950 formed five, two, three, four, and five strong interactions with D3R, respectively. However, in the representative conformations, these same ligands formed five, three, six, six, and eight strong interactions with D3R, respectively. After performing unbiased MD simulations, the numbers of strong interactions between ligands and proteins all increased except for compound D638-0102.

To more directly compare the binding stability of each ligand in both the top BPMD-scored conformation and the representative conformation, the ΔGbind value of each ligand in both complex conformations were calculated using the MM/GBSA method (Table 4). The ΔGbind values of the hit compounds in their representative conformations were all lower than that in corresponding top BPMD-scored conformations. The ΔGbind values of D638-0102, L227-1012, and F072-0950 were significantly reduced after unbiased MD simulations (with the ΔGbind values decreasing by more than 8 kcal/mol), whereas the ΔGbind values of D280-0447 and E776-0059 decreased by only 1–2 kcal/mol after unbiased MD simulations.

Summarizing the above findings, it can be concluded that both the top BPMD-scored pose and the representative pose (i.e., the ligand pose in the representative conformation after MD simulation) of each of D3R ligand were more stable than the corresponding AutoDock Vina docking poses (the ΔGbind values of which were significantly higher in each case).

Specific analysis is required to assess whether the top BPMD-scored pose or the representative pose is more likely to be the actual binding pose. The unbiased MD-simulation results showed that the positions of most ligands (including D638-0102, D280-0447, L227-1012, and F072-0950) in the representative conformation were not substantially different from the corresponding top BPMD-scored poses. These findings suggest that these top BPMD-scored poses were indeed relatively reliable from the standpoint of position preservation. However, the ligand–protein-interaction modes clearly changed after the MD simulations, and the numbers of strong ligand–protein interactions all increased after unbiased MD simulations except for compound D638-0102. In addition, after performing the unbiased MD simulations, the ΔGbind values of these D3R ligands we found were all reduced, and the ΔGbind values of D638-0102, L227-1012, and F072-0950 were significantly reduced (with ΔGbind values above 8 kcal/mol). These results suggest that the top BPMD-scored pose was not necessarily the correct pose from the standpoint of ligand–protein-binding stability.

By comprehensively analyzing changes in the ligand poses, ligand–protein interaction modes, and ΔGbind values of the hit compounds in the D3R–ligand complex before and after unbiased MD simulations, we were able to classify the compounds into three categories. Considering that the pose of D280-0447 in the representative conformation deviated only slightly from its top BPMD-scored pose, with an RMSD value of only 0.78 Å, and that the ΔGbind of D280-0447 did not decrease much after the unbiased MD simulations, we believe that the top BPMD-scored pose of D280-0447 obtained by IFD and BPMD analysis is reliable and likely to be the potential binding pose. The poses of D638-0102, L227-1012, and F072-0950 in their representative conformations did not differ markedly from their corresponding top BPMD-scored poses, with RMSD values of approximately 1.50 to 2.18 Å; however, the binding modes of these three ligands with the D3R protein changed considerably before and after unbiased MD simulations, and the ΔGbind values of D638-0102, L227-1012, and F072-0950 decreased significantly after the unbiased MD simulations (with ΔGbind values above 8 kcal/mol). Therefore, we consider that the ligand poses in the representative conformations are more plausible binding poses for D638-0102, L227-1012, and F072-0950. As for E776-0059, its position in the representative conformation deviated considerably from the position of the top BPMD-scored pose with the RMSD value reaching 3.33Å. The ligand–protein-interaction modes for E776-0059 in different conformations were also quite different, whereas the ΔGbind values of E776-0059 in different conformations are almost identical. This result shows that the binding of E776-0059 to D3R may be complicated, and different binding poses may exist with close binding energies; therefore, we could not determine which pose was most stable using the IFD-BPMD-MD-MMGBSA method.

In order to verify the reliability of the binding patterns of compounds obtained by IFD-BPMD-MD-MMGBSA method, we compared the binding patterns of our hit compounds and eticlpride with those of D3R and D2R/D3R-specific antagonistic eticlpride protein complex (PDB 3PBL) and found that they are quite different in structure, but their interaction patterns with D3R have many similarities (Figure 8 and Figure 9). First of all, eticlpride can form hydrogen bonds HB and IB with Asp110 through protonated amino groups, and this strong interaction is common in the compounds we found. Eticlpride can also form π−π stacking interaction with Phe345, which also exists in the interaction between D638-0102, L227-1012, F072-0950, and E776-0059 and D3R. Eticlpride can also form strong hydrophobic interaction with Phe346, which is similar to compound L227-1012. In addition, the kinetic simulation results of the D3R/eticlpride protein complex also show that eticlpride can have a hydrogen bond with Ser182, and this strong interaction also exists in the interaction between D638-0102, L227-1012, F072-0950, and E776-0059 and D3R. This results can verify the validity and rationality of the binding model of the hit compounds we established.

In our study, we performed unbiased MD simulations on the D3R–ligand complexes containing the top BPMD-scored poses obtained with the IFD-BPMD method to globally optimize the D3R–ligand complexes. To a certain extent, this approach could compensate for the deficiency that IFD only factors in the flexibilities of the side chains of the receptor protein during flexible docking and only optimizes the amino acid residues within 5 Å around the ligand pose when using prime for protein structure optimization. We also identified the representative conformations of MD simulations of each D3R–ligand complex and calculated the ΔGbind value of each ligand in its representative conformation using the MM/GBSA method. In this manner, not only HBs but also π−π stacking, ion bridges, and water bridges were also included when investigating the ligand–protein interactions, and a more objective indicator (the ΔGbind value) was introduced to evaluate the binding stabilities of ligands to proteins. This approach can partly compensate for the inadequacy of BPMD scores, which only focuses on the retention of positions and HBs during MD simulations and does not consider the retention of other strong interactions, such as π−π stacking and ionic bridges. We combined the unbiased MD, MM/GBSA with IFD and BPMD to develop a self-consistent and computationally efficient method for predicting and validating the binding poses of ligands to D3R. Compared with the reported IFD-BPMD method, this IFD-BPMD-MD-MM/GBSA method can provide more accurate and reliable binding poses for ligands in protein–ligand complexes.

In this study, virtual screening combined with D3R-binding affinity assays were used to identify human D3R ligands with diverse structures. In addition, we combined multiple techniques, including IFD, BPMD, unbiased MD, and MM/GBSA, into a self-consistent and computationally efficient method for predicting and validating the binding model of the hits to D3R receptors. A stepwise flow diagram of the present work is illustrated in Figure 10.

## 3. Methods and Materials

### 3.1. Structure-Based Virtual Screening

#### 3.1.1. Hardware, Software, and Online Resources

The study was carried out on a workstation with an Intel^®^ Xeon(R) Platinum 8280 L @2.26 GHz × 112 processor, 187.5 GB of RAM, an NVIDIA Corporation TU104 GPU, and a 4.5 TB hard drive running in Linux operating system. Bioinformatics software, such as AutoDock Vina [10], Schrödinger [25], and Pymol [26] and online resources, such as the National Center for Biotechnology Information [27] and *ChemDiv Database* (https://www.chemdiv.com accessed on 1 March 2019) were used in this study.

#### 3.1.2. Receptor and Ligand Preparation

The chain A of the crystal structure of D3R in complex with eticlopride (PDB code 3PBL) [28] was selected as the receptor protein for molecular docking. CHIMERA 1.5.3 software [12] was used to remove ligands, ions, and water and to minimize the protein structure, based on Gasteiger charges with 500 minimization steps. ChemDiv in-stock diverse collection database (downloaded from https://www.chemdiv.com on 1 March 2019) in structure-data file (SDF) format was selected for virtual screening. The rigid receptor and flexible ligands were parametrized using the scripts provided with the AutoDock Tools suite [29], and the parametrized data were recorded in the PDBQT file. Specifically, both the receptor and ligands were presented using a united atom model, which involved polar hydrogen atoms [30] and atomic charges estimated with the Gasteiger−Marsili method [31,32].

A receptor grid was generated with the autogrid4 program distributed with Autodock Vina software (version 1.1.2) [10]. The grid center was selected as the center of the original ligand (eticlopride), the grid spacing was set to 1 Å, and the grid dimensions were chosen so as to include all atoms of eticlopride and then augmented by ±10 Å in the x, y, and z directions. Grids were generated for each of the atom types present in the ligand set as well as for electrostatics and desolvation. 

#### 3.1.3. Validation of the Docking Method

The original ligand eticlopride was re-docked to its binding site using the AutoDock Vina software package [10] with the receptor grid generated described above. An improved empirical AutoDock Vina scoring function involving a new solvation model for organic molecules was used for docking-pose scoring [10]. The maximum energy difference between the worst and best docking modes was set to 7 kcal/mol. Molecular docking was performed using global searching-exhaustiveness settings of 8, 56, and 400, which corresponded to short, medium, and long options, respectively. The AutoDock Vina scoring function was used for docking-pose scoring. The default setting of AutoDock Vina were used for other conFigureurations. 

#### 3.1.4. Virtual Screening with AutoDock Vina and Compound Selection

Parametric ChemDiv compounds (recorded in PDBQT files) were docked to the ligand-binding site of D3R with AutoDock Vina using the short docking mode and other conFigureurations described in Section 2.1.3. The top-ranking 300 compounds with the lowest AutoDock Vina scores (docked energy) were grouped into 40 clusters based on FCFP_6 fingerprints using the “Cluster Ligands” protocol of Discovery Studio 3.0. The compound with the lowest AutoDock Vina score in each cluster was retained. Finally, 37 compounds commercially available were purchased for biological evaluations (Appendix A).

### 3.2. Biochemical Assays

#### 3.2.1. Materials

Haloperidol hydrochloride (CAS number: 52-86-8) and BP897 (N-[4-[4-(2-methoxyphenyl)-1-piperazine] butyl]-2-naphthylformamide hydrochloride (CAS number: 314776-92-6) were obtained from Biochempartner, LLC (Shanghai, China). [^3^H]-spiperone (15.2 Ci/mmol) was obtained from Amersham Biosciences (Piscataway, NJ, USA). The SureFire ERK-Phosphorylation Alpha Screen from TRG Bio-Science Pty Ltd. (Thebarton, Australia)was purchased from the distributor (PerkinElmer, Wellesley, MA, USA). The selected 37 compounds were purchased from Shanghai Topscience Co., Ltd., Shanghai, China.

#### 3.2.2. CHO–hDRD3 Cell Membrane Preparation

CHO-K1 cells expressing human D3R (CHO–hDRD3; HD Euroscreen Fast, Brussels, Belgium) were homogenized in 4× *v/w* buffer (15 mM Tris, 2 mM MgCl_2_, 0.3 mM EDTA, 1 mM EGTA, pH = 7.4 at 25 °C) with a Dounce tissue grinder and centrifuged at 40,000× *g* at 4 °C for 25 min. The supernatant was removed, and the pellet was resuspended in 4× *v/w* buffer and recentrifuged. This process was repeated two more times, and the pellet was resuspended in different buffer (composition: 75 mM Tris,12.5 mM MgCl_2_, 0.3 mM EDTA, 1 mM EGTA, 250 mM sucrose, pH = 7.4 at 25 °C) at a volume of 12.5 mL/g original weight. The preparations were then aliquoted and stored at −70 °C [11].

#### 3.2.3. [^3^H]-Spiperone-Filtration Binding Assay on Membranes from CHO–hDRD3 Cells 

The aliquoted membranes were thawed and washed once in binding buffer (50 mM Tris HCl, 5 mM MgCl_2_, 5 mM KCl, 1 mM CaCl_2_, 120 mM NaCl, and 1mM EDTA). After the wash step, the membranes were resuspended in the same buffer (3.3 mg membrane protein/assay) and incubated with 2 nM [^3^H]-spiperone raclopride in the presence or absence of a test compound (to determine the binding inhibition of the test compound or the total binding). The incubations were carried out for 120 min at 25 °C in 250 µL in individual wells of a 96-well deep-well plate. The positive-control drug BP897 was tested in parallel with the test compounds to ensure the reliability of the results. Non-specific binding was determined in the presence of 10 mM haloperidol. After incubation, the samples were filtered over UniFilter GF/B^TM^ using PerkinElmer Harvester and washed with 4 × 1 mL ice-cold binding buffer. The plate was dried at 40 °C for 1 h and 40 µL MicroScint scintillation cocktail (PerkinElmer) was added to each well. Bound radioactivity was determined in a MicroBeta 2450 microplate counter (PerkinElmer). Specific radioligand binding was defined as the difference between total binding and non-specific binding determined in the presence of excess haloperidol. Inhibition percentage (%) = (total binding counts − test compounds binding counts)/(total binding counts—non-specific binding counts) [11]. The concentration of compound required to inhibit specific binding by 50% (i.e., the IC_50_ value) was determined based on the concentration-dependent displacement curves and sigmoidal curve fitting.

### 3.3. Exploring the D3R-Binding Modes of Hit Compounds via Induced-Fit Docking (IFD), Binding Pose Metadynamics Simulation, Unbiased Molecular Dynamics (MD) Simulation, and Molecular Mechanics Generalized Born-Surface Area (MM/GBSA) Analysis

#### 3.3.1. IFD Analysis to Determine Potential Ligand-Binding Poses with D3R

Flexible docking was performed using the IFD protocol [19], as implemented in the Schrödinger suite. The prepared D3R structure (PDB 3PBL) was used as the receptor. A 15 × 15 × 15 Å^3^ grid centered on the ligand, eticlopride, was selected as the docking space. The following standard IFD sampling protocol was used. First, the protomers/stereoisomers of ligands generated in the ligand-preparation step were initially docked using Glide [33], the receptor van der Waals radii scaling was set at 0.70, and the ligand van der Waals radii were scaled to 0.5 to soften the potentials. Then, sampling and minimization of the sidechain positions of the binding-site residues within 5 Å of the docked ligand were performed using Prime [34], followed by re-docking using Glide. Finally, the binding energy for each output pose was estimated as the IFD score.

#### 3.3.2. Chemoinformatics: Structural-Interaction Fingerprints

Output IFD poses for each of hit compound were clustered using Interaction Fingerprints (SIFt) contact-similarity scoring [21], as implemented in the Schrödinger suite, to exclude redundant poses. The default options were used to generate the fingerprints of the IFD poses. For each ligand, including its protomers/stereoisomers, the poses with the top IFD score of each cluster were grouped. For each group, the top three ranked IFD poses were selected for binding pose metadynamics (BPMD) simulations.

#### 3.3.3. BPMD Simulations

Ligand–receptor complexes containing the selected IFD output poses were used as input complexes. The complexes were embedded in a 1-palmitoyl-2-oleoylphosphatidylcholine (POPC) bilayer, solvated in an orthorhombic box with a 10 Å buffer in each dimension consisting of simple point-charge (SPC) water molecules and 0.15 M NaCl and neutralized with chloride ions using System Builder in Schrödinger Suite 2019. BPMD simulations were performed using the Bind Pose Metadynamics protocol, implemented in the Schrödinger Suite with default parameters [20]. A collective variable (CV) was defined as the RMSD of a ligand’s heavy atoms relative to their starting positions in the IFD output pose. Metadynamics simulations were carried out for 10 ns, and frames with 200 ps interval were recorded. Three scoring functions, including the persistence score (PersScore), composite score (CompScore), and pose score (PoseScore) were used to rearrange poses [20].

#### 3.3.4. Unbiased MD Simulations

Desmond [22] was used to perform MD simulations for protein–ligand complexes containing the top metadynamics-ranked poses. The dynamics system was set up using the system-builder module as described in Section 3.3.3. Initially, the system was relaxed using the Desmond relaxation model. The completed equilibration run was followed by a production run performed under normal temperature and pressure conditions (300 K and 1.103 bar, respectively), using isothermal-isobaric (NPT) ensemble and particle mesh Ewald (PME) electrostatics with a cutoff of 9 Å. Time-step calculations were performed every 2 fs. The simulation job was carried out over a period of 100 ns. RMSD and root-mean-square fluctuation (RMSF) values were calculated and analyzed using Simulation Interaction Diagrams in Desmond.

#### 3.3.5. Interactions Analysis, Trajectory Clustering, and MM/GBSA Calculations

The Trajectory Frame Clustering tool in Maestro was used to estimate the most populated representative structure for each MD simulation. The backbone atoms were set for the RMSD matrix calculations using trajectory clustering. Ten frames were used for the trajectory-frame extraction interval, 1000 frames were used when clustering each trajectory, and the maximum output number of clusters was set to 10. The structure with the largest number of neighbors in the structural cluster was selected as the representative conformation, which was analyzed by performing binding free energy (ΔGbind) calculations using the Prime MM/GBSA tool in Maestro. The VSGB solvation model and OPLS3e force field were set for the ΔGbind calculations.

## 4. Conclusions

D3R was selected as the target, and 1,535,478 compounds in *ChemDiv* database were virtually screened by AutoDock Vina docking method. The 300 molecules with the lowest AutoDock Vina score were retained for fingerprint cluster analysis. Thirty-seven commercial compounds were selected for biological evaluation. Among them, 27 compounds showed more than 50% D3R inhibition at the concentration of 10 μM, and 23 compounds showed more than 70% D3R inhibition at the concentration of 10 μM. Thirteen compounds showed more than 80% inhibition of D3R at the concentration of 10 μM, six compounds showed more than 90% inhibition of D3R at the concentration of 10 μM, and IC_50_ values were measured, among which five compounds had inhibitory effect on D3R receptor, with IC_50_ values ranging from 0.97 μM to 1.49 μM.

These hit compounds show good structural diversity, but the BPMD simulation method shows that the docking poses obtained by docking the hit compounds with AutoDock Vina are not accurate enough. The method combine IFD with BPMD was used to explore the potential binding forms of ligands bound to D3R. The results show that this combined IFD-BPMD method can indeed determine the approximate binding position of D3R ligand, but it cannot accurately predict the binding mode of ligand. This is due to some algorithm defects of FD and BPMD. In this work, we used the unbiased MD simulation to globally optimize the protein complex containing the ligand poses with the highest BPMD score and included π-π stacking, ion bridge, and water bridge in the study of ligand–protein interaction. A more objective index (ΔGbind value) is also added to evaluate the stability of ligand binding. By combining unbiased MD, MM/GBSA with IFD and BPMD, a self-consistent and computationally efficient method for predicting and verifying the binding of ligand to D3R poses was obtained. Compared with the reported IFD-BPMD method, this IFD-BPMD-MD-MM/GBSA method can provide more accurate and reliable binding poses for ligands in protein–ligand complexes.

## Figures and Tables

**Figure 1 molecules-28-00527-f001:**
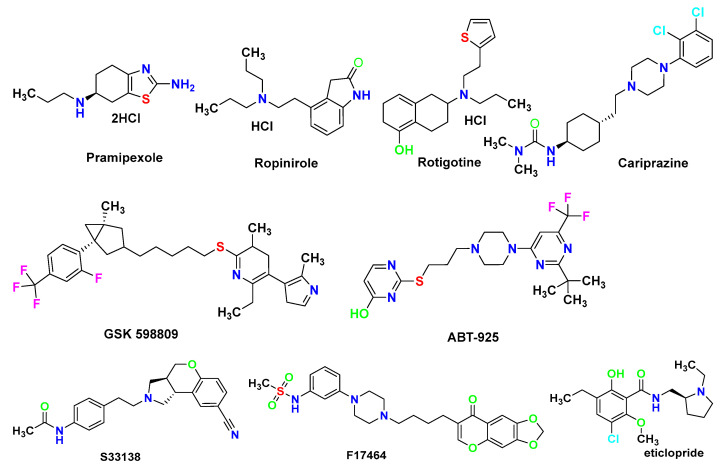
Structures of several discovered D3R-preferring compounds.

**Figure 2 molecules-28-00527-f002:**
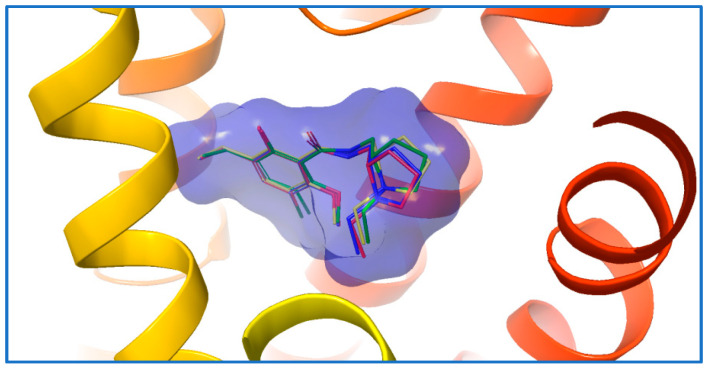
Comparison of the output poses of Eticlopride in short, medium, and long search modes of AutoDock Vina with the original pose in the crystal. The original pose is presented as thick, green tubes, and the predicted poses for short, medium, and long search modes are presented as blue, yellow, and pink thin tubes, respectively.

**Figure 3 molecules-28-00527-f003:**
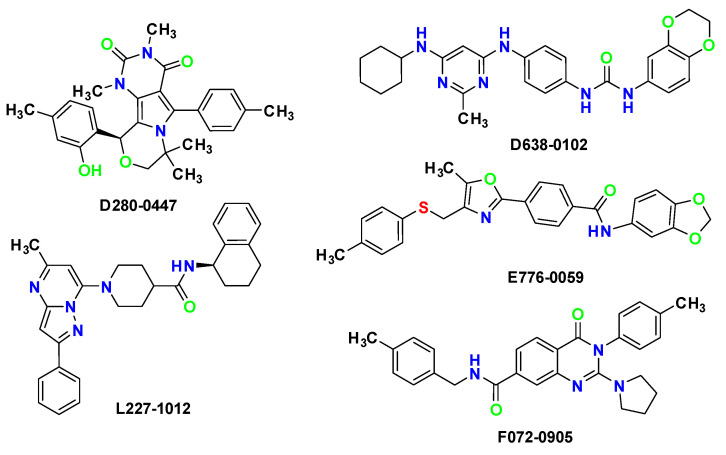
The structures of the five most active hit compounds.

**Figure 4 molecules-28-00527-f004:**
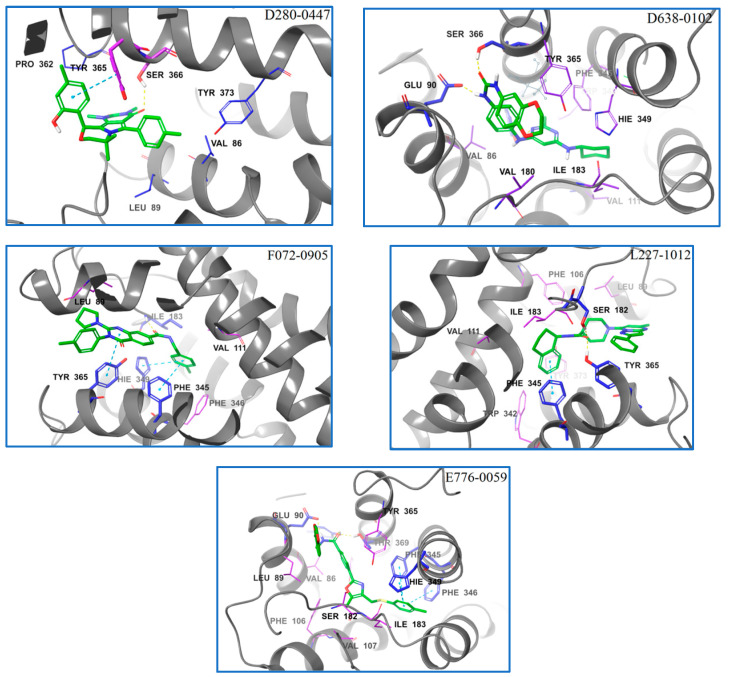
The protein–ligand interactions diagrams of the AutoDock Vina pose of the hit compounds binding to D3R proteins. In the protein–ligand interaction diagrams, the D3R protein backbones are represented by gray cartoons; ligands by thick, green tubes; residues that have strong interactions with ligands are presented by thick plum tubes; residues that have hydrophobic interactions with the ligand are represented by thin, blue tubes.

**Figure 5 molecules-28-00527-f005:**
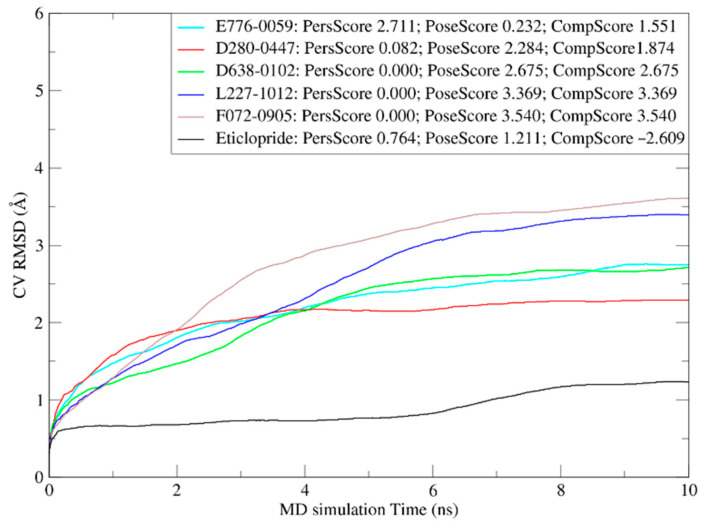
Plots of the RMSD estimate averaged over all 10 trials versus the simulation time for the BPMD runs of DD3R/eticlopride complex and AutoDock Vina docking poses of hit compounds. The PoseScore, PersScore, and CompScore values for the these D3R–ligand complexes are shown in the legend.

**Figure 6 molecules-28-00527-f006:**
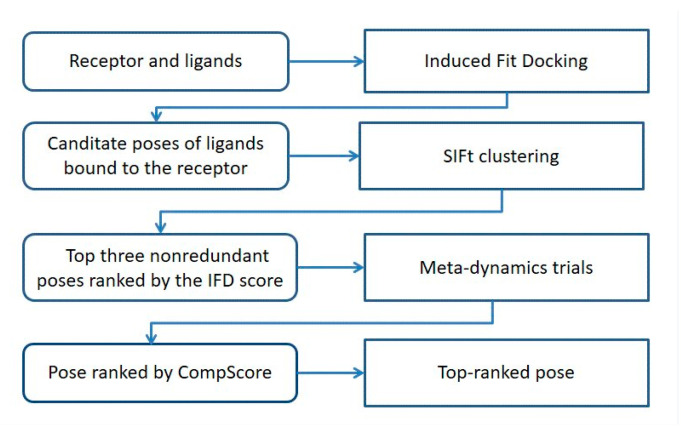
Schematic representation of the procedures used to explore the potential binding poses, based on a combination of IFD and BPMD analysis.

**Figure 7 molecules-28-00527-f007:**
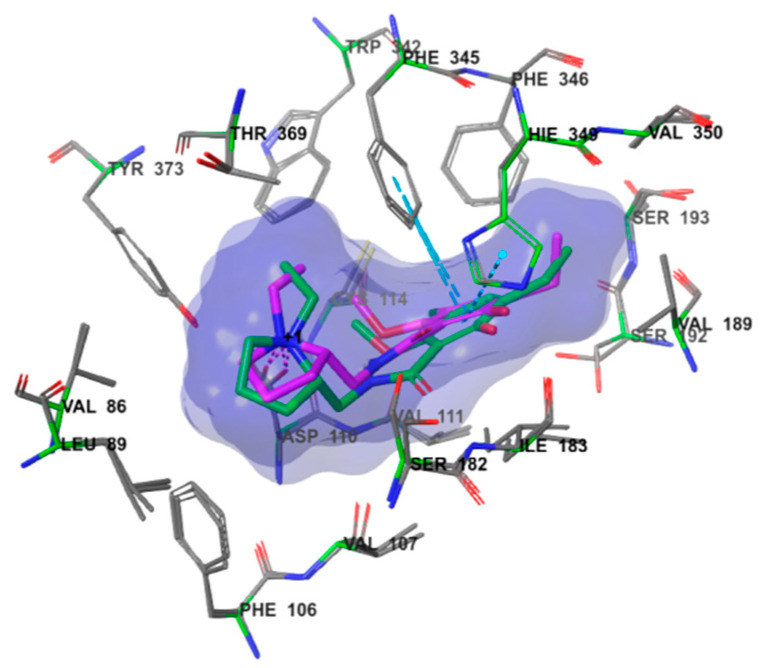
The comparation of top BPMD-scored pose with the original pose in PDB 3PBL of eticlopride.

**Figure 8 molecules-28-00527-f008:**
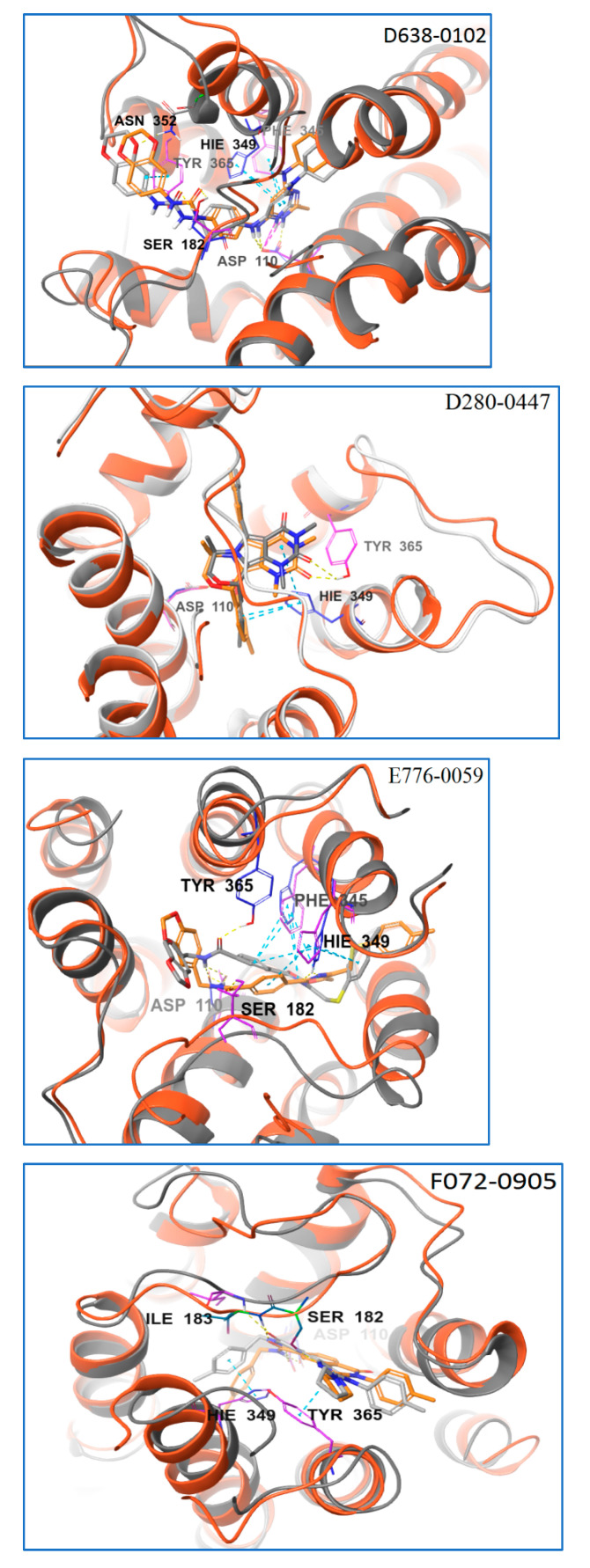
Alignment of the top BPMD-scored conformations with representative conformations of D3R–ligand complex and the binding pose of eticlopride in PDB 3PBL. In the top BPMD-scored conformations, the D3R protein backbones are represented with a red-orange color; the ligands are represented by thick, orange tubes; and the residues are represented by thin, plum-colored tubes. In the representative conformations, the D3R protein backbones are represented with a gray color; the ligands are represented by thick, light-gray tube; and the residues are represented by thin, blue tubes. HBs are represented by yellow dashes, π−π stacking is represented by light-blue dashes, and ionic bridges are represented by red-orange dashes.

**Figure 9 molecules-28-00527-f009:**
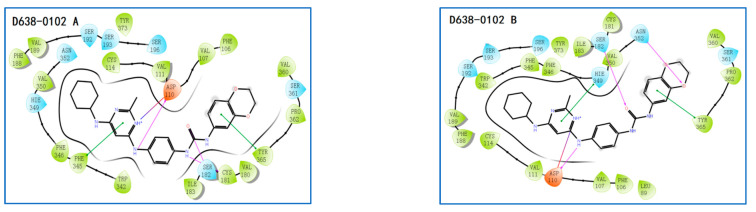
Schematic 2D diagrams of the protein–ligand interactions of the top BPMD-scored conformations (**A**) and representative conformations (**B**) of different D3R–ligand complexes and the schematic 2D diagrams of the D3R–eticlopride interactions of PDB 3PBL. HBs are represented by plum-colored arrows; π−π stackings are represented by dark-green lines.

**Figure 10 molecules-28-00527-f010:**
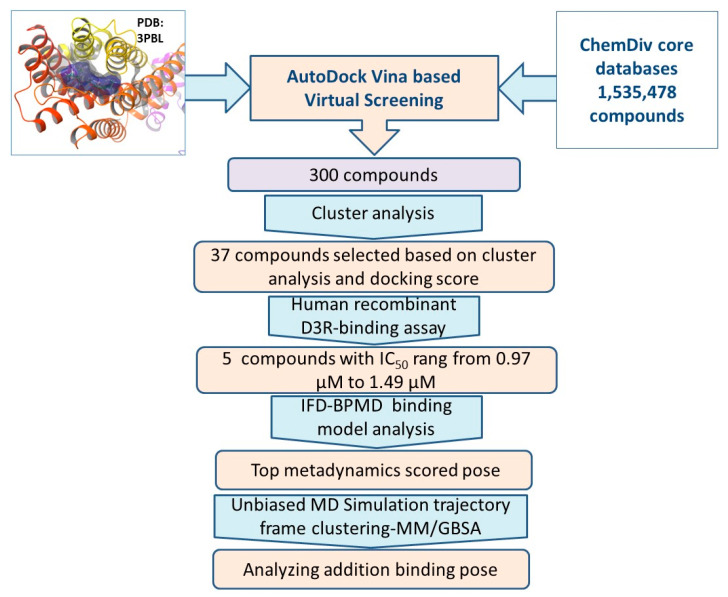
Flow chart of the work performed in the present study.

**Table 1 molecules-28-00527-t001:** The serial number, AutoDock Vina score, and experimental binding affinities of selected compounds.

	Compound	AutoDock Vina Score (kcal/mol)	Inhibition Rate at 10 µM	IC_50_ (μM)		Compound	AutoDock Vina Score (kcal/mol)	Inhibition Rate at 10 µM (%)	IC_50_ (μM)
	BP897		≥98.0%						
1	8017-6887	−13.0	74.8		15	F366-0225	−13.4	80.1	5.98
2	8018-0047	−12.4	79.1		16	F366-0245	−12.6	91.1	3.45
3	C645-0112	−11.8	82.4	4.51	17	F486-0373	−12.0	71.7	
4	C736-0093	−12.5	67.8		18	G373-0280	−11.8	74.8	
5	D063-1105	−13.3	78.6		19	G435-0137	−14.0	75.7	
6	D122-0034	−13.1	84.1	4.52	20	K306-0682	−13.6	84.2	4.53
7	D122-0078	−12.8	82.3	4.49	21	L100-0151	−13.9	71.3	
8	D638-0102	−12.0	93.6	1.48	22	L112-0768	−13.3	66.4	
9	D280-0447	−12.2	99.3	1.25	23	L153-0098	−12.3	78.4	
10	E776-0059	−11.8	96.9	0.97	24	L227-1012	−12.5	94.8	1.49
11	E776-1501	−12.5	61.1		25	L759-0276	−13.8	63.4	
12	E859-1320	−13.1	77.6		26	L759-0287	−13.2	89.3	4.11
13	F072-0905	−12.4	99.5	1.41	27	G544-1316	−12.9	87.5	4.35
14	F351-0364	−13.5	71.2						

**Table 2 molecules-28-00527-t002:** Poses clustering based on interaction fingerprints.

Compounds ID	Poses	Grouped into Clusters
Eticlopride	31	7
D638-0102	27	15
D280-0447	8	4
E776-0059	2	2
F072-0905	14	4
L227-1012	2	2

**Table 3 molecules-28-00527-t003:** The selected top three IFD-scored poses and their BPMD simulation: PoseScore, PersScore, and CompScore.

Compounds ID	Pose1				Pose2				Pose3			
	IFD Score	Pose Score	Pers Score	Comp Score	IFD Score	Pose Score	Pers Score	Comp Score	Ifd Score	Pose Score	Pers Score	Comp Score
Eticlopride	−474.94	1.008	0.336	−0.672	−474.79	0.944	0.477	−1.441	−474.70	1.894	0.723	−1.721
D638-0102	−865.42	1.666	0.752	−2.094	−867.53	1.551	0.503	−0.964	−864.91	2.286	0.409	0.241
D280-0447	−855.76	1.834	0.364	0.014	−855.46	1.865	0.241	0.66	−856.59	1.355	0.00	1.355
F072-0905	−856.49	2.107	0.045	2.107	−855.12	2.129	0.00	2.129	−855.02	1.913	0.373	0.048
L227-1012	−867.86	1.835	0.352	0.758	−867.30	2.117	0.00	2.117	-	-	-	-
E776-0059	−859.95	2.246	0.527	−0.389	−856.17	1.815	0.00	1.815	-	-	-	-

**Table 4 molecules-28-00527-t004:** Key ligand–protein interactions and ΔGbind values of ligands in top BPMD-scored conformations and representative conformations as well as the RMSD values for the ligand poses in the top BPMD-scored conformations and representative conformations.

	D638-0102		D280-0447		L227-1012		F072-0950		E776-0059	
	①	②	①	②	①	②	①	②	①	②
Asp110	HB(A) IB	HB(A) IB	HB(A)	HB(A)	HB(A)	HB(A)	HB(A)	HB(A)	HB(A)	HB(A)
Ser182	HB(D) HB(A)	HB(D)			HB(D)		HB(D)		HB(D)	
Ile183					HB(D)	HB(D)	HB(D)	HB(D)		
Phe345	π−π					π−π	π−π	2*π−π	π−π	2*π−π
Phe346						π−π				
Hie349		π−π		2*π−π			π−π	2*π−π HB(A)	π−π	2*π−π
Asn352		HB(A)								
Tyr365	π−π	π−π	HB(D)			HB(D) π−π		HB(D)		HB(D)
ΔGbind	−82.45	−90.35	−70.59	−73.04	−97.84	−118.15	−81.32	−91.97	−98.90	−99.42
RMSD	1.50		0.78		1.60		2.18		3.33	

①Top BPMD-scored conformation; ② representative conformation; HB(A), HB acceptor; HB(D), HB donor; π−π, π−π stacking; IB, ionic bridge.

## Data Availability

Not applicable.

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
