# Peer review of "Identifying Dopamine D3 Receptor Ligands through Virtual Screening and Exploring the Binding Modes of Hit Compounds"

_molecules, 2023, doi:10.3390/molecules28020527_

Round 1

Reviewer 1 Report

The dopamine D3 receptor is an important central nervous system target for treating various neurological diseases. Due to the requirement of novel ideal D3R antagonist, the authors carried out the research of the area in the paper.

The Autodock Vina docking method was used to virtually screen the 1535478 compounds in ChemDiv database. 37 commercial compounds were selected from the screened 300 molecules for biological evaluation. Five compounds with the highest inhibition rate were selected to further study the binding mode with the help of IFD-BPMD and MM/GBSA prediction and analysis.

The method is meaningful to screen some novel D3R ligands. Hence, the manuscript is suitable to be published by this journal. But the following revision is still required.

1. Cariprazine is a known agent targeting the D3R, can it help to verify the validity or rationality of the established binding model in the manuscript? Please add the information in the manuscript.

2. The PDF layout of the manuscript seems to be inadequately edited, which caused some editorial problems, such as the unified size of tables or figures (Figure 4, 8 and 9, etc).

3. What are the advantages of this IFD-BPMD-MD-MM/GBSA method over the reported IFD-BPMD method? Please give proper explanation.

Author Response

  1. Cariprazine is a known agent targeting the D3R, can it help to verify the validity or rationality of the established binding model in the manuscript? Please add the information in the manuscript.

Response: Thanks for the constructive suggestion. Cariprazine a D3R-preferred D3R/D2R partial agonist. In this paper, Cariprazine is only used as a positive control for detection. The currently known SAR information of Cariprazine and D3R is not much and fragmentary. In fact, the D3R protein crystal structure used in the virtual screening in this paper contains a D2R/D3R-specific antagonist eticlopride. We feel that it is more direct and accurate to directly use the binding model of eticlopride to verify the validity or rationality of the established binding model in the manuscript. This article has supplemented relevant content in the main text.

The PDF layout of the manuscript seems to be inadequately edited, which caused some editorial problems, such as the unified size of tables or figures (Figure 4, 8 and 9, etc).

Response: We are sincerely grateful for your positive comments and helpful suggestions. You mentioned that the  size of tables or figures and other pictures have been modified in the article.

  1. What are the advantages of this IFD-BPMD-MD-MM/GBSA method over the reported IFD-BPMD method? Please give proper explanation.

Response: Thank you so much for your important comment and question. On page 18 of this article, we have explained the advantages of this IFD-BPMD-MD-MM/GBSA method over the reported IFD-BPMD method. For details, please see the part marked in red below (see also page 18 of the main text):

In our study, we performed unbiased MD simulations on the D3R–ligand complexes containing the top BPMD-scored poses obtained with the IFD-BPMD method to globally optimize the D3R–ligand complexes. To a certain extent, this approach could compensate for the deficiency that IFD only factors in the flexibilities of the side chains of the receptor protein during flexible docking and only optimizes the amino acid residues within 5 Å around the ligand pose when using Prime for protein structure optimization. We also identified the representative conformations of MD simulations of each D3R–ligand com-plex and calculated the ΔGbind value of each ligand in its representative conformation using the MM/GBSA method. In this manner, not only HBs, but also π−π stacking, ion bridges, and water bridges were also included when investigating the ligand–protein in-teractions, and a more objective indicator (the ΔGbind value) was introduced to evaluate the binding stabilities of ligands to proteins. This approach can partly compensate for the inadequacy of BPMD scores, which only focuses on the retention of positions and HBs during MD simulations and does not consider the retention of other strong interactions, such as π-π stacking and ionic bridges. We combined the unbiased MD, MM/GBSA with IFD and BPMD to develop a self-consistent and computationally efficient method for predicting and validating the binding poses of ligands to D3R. (Therefore we can draw the following conclusions:)Compared with the reported IFD-BPMD method, this IFD-BPMD-MD-MM/GBSA method can provide more accurate and reliable binding poses for ligands in protein-ligand complexes.

Reviewer 2 Report

Authors have will written the manuscript, only the concern with figure presentation.

1. All the figures should be clear with high resolution.

2. RMSD and RMSF graphs provided in the supplementary files, should be provided in merged form so that readers can see the difference in RMSD and RMSF very easily. Authors can use the .dat files generated from simulation interaction diagram or they can recreate the csv files for RMSD or RMSF from Simulation event analysis.

3. Authors have not mentioned number of SPC used to solvate each system.

4. MD simulation methodology seems to be incomplete, as they have not provided thermostat and barostat methods used in this study.

Author Response

  1. All the figures should be clear with high resolution.

Response: We are sincerely grateful for your positive comments and helpful suggestions. The figures and other pictures have been modified in the article.

  1. RMSD and RMSF graphs provided in the supplementary files, should be provided in merged form so that readers can see the difference in RMSD and RMSF very easily. Authors can use the .dat files generated from simulation interaction diagram or they can recreate the csv files for RMSD or RMSF from Simulation event analysis.

Response: We are sincerely grateful for your positive comment and helpful suggestion. Both the RMSD and RMSF diagrams presented in this paper were automatically generated from the "out.cms" file generated by the MD simulations using the "Simulation Interaction Diagram" tool in the Desmond MD package. We have no idea how to merge these auto-generated files into one file. In addition, we feel that even if RMSD and RMSF graphs are still provided to readers in the current form, it will not seriously affect readers see the difference in RMSD and RMSF. So please allow us not to deal with these RMSD and RMSF graphs.

  1. Authors have not mentioned number of SPC used to solvate each system.

Response: Thanks for your constructive suggestion. In this paper, when performing BPMD on the "ligand–receptor complexes containing the selected IFD output poses” and Unbiased MD simulations on the " protein–ligand complexes containing the top metadynamics-ranked poses", we used the “System Builder” protocol twice to set up the biological systems for simulations with Desmond. Theoretically speaking, for a given protein-ligand complex, if the charge state of protein and ligand, the way of introducing membrane, the shape and size of Box are all fixed, then the number of SPC in the simulation system given by "System Builder" " should be fixed. However, even for the same protein-ligand complex, "the ligand-receptor complex containing the selected IFD output pose" and "the protein-ligand complex containing top metadynamics-ranked poses “are two different conformations. For these two different conformations, even if the charge states of proteins and ligands, the introduction method of membrane proteins, and the shape and size of Box are all fixed, the number of SPCs in the simulation system given by "System Builder" will be slightly different. Therefore, we did not mention the number of SPC used to solvate each system in our paper.

In addition, theoretically speaking, parameters such as the charge of proteins and ligands, the way of introducing them into the membrane, and the shape and size of Boxes are more important when constructing simulation systems. How many water molecules are introduced is derived from these parameters and has no significant effect on the simulation, so in general it is not necessary to report the number of water molecules as a parameter when doing molecular dynamics simulations with Desmond.

  1. MD simulation methodology seems to be incomplete, as they have not provided thermostat and barostat methods used in this study.

Response:Thanks to the reviewers for your valuable comments. The Desmond we use is a semi-automated explicit solvent molecular dynamics program. When doing a regular NPT simulation, a protocol called "Relax model system before simulation" will be executed by default, which allows you to relax the model system before performing the simulation. A series of minimizations and short molecular dynamics simulations are performed to relax the model system before performing the simulation you set up.